# Potentials and pitfalls of increasing prosocial behavior and self-efficacy over time using an online personalized platform

**Sharon T. Steinemann**◯*, **Benjamin J. Geelan, Stephan Zaehringer, Kamalatharsi Mutuura, Ewgenij Wolkow**[1], **Lars Frasseck, Klaus Opwis**

Department for General Psychology and Methodology, Faculty of Psychology, University of Basel, Basel, Basel-Stadt, Switzerland

* sharon.steinemann@unibas.ch

## Abstract

### Background

This longitudinal mixed methods experimental study aimed to better understand the interplay between digital technology exposure over time, self-efficacy, and prosocial behavior in everyday contexts.

### Methods

66 psychology students tracked their daily prosocial behavior over three weeks. Additionally, half of the participants were randomly assigned to receive access to an online platform, which made personalized suggestions for prosocial actions to complete. Qualitative post-study interviews complemented quantitative measures.

### Results

Platform exposure had no measurable impact beyond that of tracking over time on either prosocial behavior or self-efficacy. Tracking increased self-efficacy to perform everyday prosocial actions, but did not affect self-efficacy to impact change. Prosocial behavior was predicted by self-efficacy to impact change. Enjoyment of the platform predicted completing higher numbers of suggested prosocial actions and was related to a higher likelihood to continue using the platform in the future. Avenues for increasing platform effectiveness include context-specific action personalization, an effective reminder system, and better support for the development of self-efficacy to impact change through meaningful actions.

### Conclusion

Technology for prosocial behavior should be enjoyable, capable of being seamlessly integrated into everyday life, and ensure that suggested actions are perceived as meaningful in order to support the sustainable development of self-efficacy and prosocial behavior over time.

**Data Availability Statement:** The data can be accessed on the Open Science Framework using the following link: https://osf.io/dnu3y/.

**Funding:** STS was supported by the Freie Akademische Gesellschaft Basel (https://www.fag-basel.ch/). The funder played no role in study design, data collection and analysis, decision to publish, or preparation of manuscript.

**Competing interests:** The authors have declared that no competing interests exist.

# Introduction

Advances in digital technology bring with them an exceptional potential for beneficial impacts on individuals, communities, and, ultimately, societies. Technology can inform, connect, encourage, and facilitate action on a scale previously impossible [1, 2]. At its best, technology can support people throughout their lives; Assisting them in meeting their goals, living healthier lives, increasing wellbeing, helping others, and contributing to their community (e.g., [3–6]). However, at its worst, both through unintended consequences and by design, technology can also facilitate hostility and distress and offer platforms for ridicule and anger (e.g., [7, 8]). To guide the development of technology towards benefit and not to harm, a deeper understanding is necessary of the interactions between new forms of technology in everyday contexts and human behavior. Without this understanding, any attempts at creating innovative solutions run the risk of being ineffectual, or worse, damaging. Research on the impact of technology on reaching goals and supporting healthy behavior has been the focus of several recent studies [5, 6, 9]. However, relatively little research has focused on supporting prosocial behavior directed towards others, such as everyday acts of helping friends and strangers [1].

The aim of this exploratory study is therefore to examine the prospects of using an early iteration of an online platform to support prosocial behavior in everyday contexts and increase self-efficacy, an important predictor of behavior [10], in its users over time. The following sections will give an overview of the theoretical foundation on which this work builds.

## Theoretical background

**Behavior change and self-efficacy.** Self-efficacy is the belief in one's abilities to successfully perform the actions necessary to achieve a specific goal [10]. Self-efficacy theory postulates that increasing self-efficacy for a specific behavior (e.g., biking to work), will increase the likelihood that this behavior will be undertaken. This close relation between self-efficacy and behavior has been supported by a plethora of empirical evidence across different areas, such as educational outcomes [11], disease prevention [12], and pro-environmental behavior [13]. Self-efficacy beliefs can vary in terms of their generality, from task-specific self-efficacy beliefs to general self-efficacy beliefs which transcend across tasks and situations [14].

However, the predictive power of self-efficacy on behavior tends to be greatest, the more specifically self-efficacy is measured [10]. For example, one's belief in one's ability to bike to work should better predict the probability of actually biking to work than the more general belief in one's overall competence to perform moderate exercise activities.

To increase self-efficacy, mastery experiences, defined as experiences in which one has successfully performed actions relevant to one's ultimate goal, have been found to be particularly effective [15]. So, for example, one's belief in one's ability to successfully bike to work will likely be increased by having experienced biking successfully in different levels of traffic, in different types of weather, and at different times of day.

Research on the relationship between self-efficacy and prosocial behavior has found general social self-efficacy (one's general belief in one's ability to interact well with others) to predict public prosocial behaviors, but not anonymous prosocial behavior [16]. Examining a more specific form of self-efficacy, White et al. [17] found higher belief in one's ability to have an impact on an observed social injustice to increase participants' intentions to purchase fair trade products. Importantly, this self-efficacy to impact social change, and ultimately the intention to purchase fair trade products, was predicted by how effective participants were led to believe the fair trade solution to be.

This highlights the importance of mastery experiences for both prosocial self-efficacy and ultimately for probability of repeated prosocial behavior. This is also supported by past

research by Sargeant et al. [18] on the importance of feeling accomplished for people contributing to charities.

This is also in line with research on charitable giving where people have been found to be more likely to donate if they feel as though their contribution will make a difference [19, 20]). Considering this, we argue for the importance of the self-efficacy belief we define as *change impact self-efficacy*, or, the belief in one's abilities to successfully perform actions that will lead to prosocial change. This concept is the generalization to prosocial change of justice restoration self-efficacy, as measured by White et al. [17] in the context of fair trade behavior.

Change impact self-efficacy is however still a fairly broad concept, considering that prosocial change can be achieved in a variety of ways. As our goal is to understand everyday prosocial behavior, ideally we would therefore wish to examine *everyday helping self-efficacy* or, belief in one's ability to perform everyday actions, which are intended to help others. As no such measure so far exists, a first questionnaire for everyday self-efficacy was developed in this study.

**Changing behavior through technology.** The question of how technology can positively impact behavior has been of great interest in the past decades. Past studies have focused on changing behavior such as healthy eating (e.g., [21]), improved learning outcomes [22], supported smoking cessation [23–25] and increased physical activity (e.g., [26, 27]), wellbeing [4], and stress relief [4].

These technologies leverage the unique potentials of technology-based solutions to support their users in performing behavior they may not otherwise have been motivated or capable of doing [1, 28, 29]. For example, exercise games can lead to exercise activities being experienced as more enjoyable, leading to users engaging for longer and ultimately burning more calories than with non-game equivalents [26]. Similarly, applications to assist in learning languages, such as *Duolingo*, can encourage users to learn languages on their own outside of more traditional classroom settings [30].

Recent reviews point to a multitude of specific features of digital technology, which may be particularly helpful in changing behavior. Self-Monitoring and -tracking, personalized feedback, social sharing, reminders, as well as gamified elements including points, progress visualization, and virtual rewards are features frequently used to support behavior change [5, 6, 31]. Many of these features may be particularly effective due to the fact that they support mastery experiences by making users aware of their progress and successes and prompting them to improve slowly over time.

Indeed, studies on the effectiveness of these features indicate that particularly tracking, feedback, and gamified elements are in many cases able to affect behavior [6, 31]. However, findings are not always clear-cut, demonstrated by the conclusions of a recent systematic literature review examining gamified elements, which noted that particularly rigorous studies have been more likely to find mixed or non-significant results on the effectiveness of such features [31]. Thorough investigations under controlled conditions are therefore necessary in order to draw consistent conclusions [31]. Several avenues for improvement are open here, as research on the impact of technology on behavior change has been hampered in the past by a number of theoretical and methodological issues. Firstly, there is an over reliance on tracking of behavior as a way of facilitating behavior change [32]. Secondly, the tools examined often lack theoretical grounding for why they are expected to work, making it difficult to answer the question of why tools may or may not have produced the expected impact [5, 32]. Thirdly, a lack of experimental manipulation makes conclusions as to the effectiveness of tools difficult [31]. Relatedly, as often several features are combined in one tool, it becomes difficult to pinpoint the specific effect of individual features [5, 31]. And finally, there is a dearth of studies examining effects over time [5, 6, 28, 31], thereby leaving it unclear to which extent technology allows

for sustainable behavior change over longer periods of time, and which studies are illustrating novelty biases through the introduction of new technologies.

**Changing prosocial behavior through technology.** Prosocial behaviors are defined as actions, which are intended to benefit one person or multiple people other than oneself [33, 34]. It covers behavior such as helping a stranger by giving up one's seat in public transport, comforting a friend by cooking them dinner, or taking a dog from a rescue shelter for a walk. Next to benefits to others, prosocial behavior is also related to increases in benefactor wellbeing [35–39]. Despite the breadth of behaviors included in this definition and the benefits to all parties involved, research examining the impact of technology on such behavior is scarce. What research there is however, highlights the potential of technology to impact prosocial behavior [3, 40]. For example, research on the impact of games on prosocial behavior has found that playing as a refugee in a game can lead to players donating higher amounts to aid refugees directly after the game [41]. Relatedly, research on virtual reality applications found virtually simulating what it is like to be color blind to lead to increased volunteering to help color blind individuals [42] and virtually simulating flying like Superman leading participants to be more likely to help out the experimenter at the end of the study by picking up dropped pens [43]. Outside of the lab setting, a study by Konrath et al. [44] found that sending high-empathy text messages for two weeks can be enough to increase prosocial behavior in text-message recipients. While these series of results are encouraging, the study by Konrath et al. [44] alone offers insight into whether prosocial behavior can be affected over time and in settings beyond the immediate study situation. Further examination of the sustainable long-term benefits of technology on prosocial behavior is therefore needed.

**Aim of this study.** Based on this past research, the intent of this study was to use an experiment to explore the potential of technology to positively affect prosocial behavior and self-efficacy over time.

On the foundation of self-efficacy theory literature, our goal was to create a platform, with which participants could practice successfully mastering prosocial actions, thereby increasing self-efficacy and ultimately their prevalence of prosocial behavior. To facilitate these mastery experiences within the platform, we used the COM-B (Capability, Opportunity, Motivation, Behavior) model developed by Michie et al. [29]. This model was developed based on an extensive review of behavior change theory literature and argues that behavior (e.g. going for a 30-minute jog) is facilitated when one is capable of performing an action (e.g., one can keep up a good running pace for 30 minutes), has the opportunity to do so (e.g one has 30 minutes to spare and has a running trail close by), and is motivated to do so (e.g. one enjoys running; or would like to improve one's fitness). While several similar models exist, the COM-B model was chosen, due to having a careful foundation on behavior change theories and being well-established in the field of applied behavior change research (e.g. [26, 45]).

Following this model, we created a digital platform, which allowed prosocial action suggestions to be personalized to individual participants' capabilities, opportunities, and motivations. The platform also tracked the number of suggested actions participants reported to have performed in any given week. In addition to offering personalized actions, the platform was designed to include gamified elements in form of progress visualization (a map showing the location of their performed actions) and virtual rewards in the form of badges for actions completed (see Materials section for details).

At the same time, participants were asked to track their daily prosocial behavior, which they had performed external from and without the prompting of the platform. These features were chosen as a first set, which based on past research offered promise of supporting behavior change.

**Table 1.**

| Features | Tracking-Only Group (TG) | Platform Group (PG) |
|---|---|---|
| Tracking everyday prosocial behavior | X | X |
| Personalized suggested actions | - | X |
| Progress Visualization | - | X |
| Virtual Rewards | - | X |

Overview of the difference in feature exposure between the two experimental groups. The Tracking-Only Group simply received daily reminders to track their everyday prosocial behavior. The Platform Group additionally received personalized action suggestions, progress visualization and virtual rewards based on the actions they performed.

To examine the effect of the additional gamified platform features compared to merely tracking behavior on the platform's effectiveness, we created two experimental conditions. Participants in the first condition (Platform Group) used the platform while tracking their daily prosocial behavior. Participants in the second condition (Tracking-Only Group) merely tracked their daily prosocial behavior, without being exposed to any of the other features available on the platform (personalized suggested action, progress visualization, and virtual rewards). An overview of the features each group was exposed to can be seen in Table 1.

To understand the effectiveness of these different features over time, we conducted a three week longitudinal study, in which we had participants track their daily prosocial behavior, suggested actions completed, and self-efficacy. Past research on games, gamification, and other media experiences over time has highlighted the particular importance of enjoyment and other forms of gratification for users to repeatedly engage with technology [46–48]. We therefore wished to examine the extent to which interacting with the platform was experienced as enjoyable [49] and its effect on engaging with the platform through the completion of suggested actions. Finally, we wanted to ensure that we would understand not only what actions people were doing, but also the reasons behind their actions. Therefore we planned to include a qualitative interview at the end of the study, which would provide a richer understanding of how participants had experienced the tracking and the platform, as well as ways in which both could be improved upon in the future.

Of interest to us was the exploration of the following research questions.

**RQ1: Daily prosocial behavior & platform exposure**
Over the course of the three weeks. . .

(a) Does exposure to the platform features (Platform Group) affect daily prosocial behavior differently to merely tracking daily prosocial behavior (Tracking-Only Group)?

(b) Is daily prosocial behavior related to self-efficacy?

**RQ2: Suggested action completion using the platform**
Over the course of three weeks. . .

(a) Is completion of suggested actions related to self-efficacy?

(b) Is completion of suggested actions related to enjoyment of the platform?

**RQ3: Self-efficacy & platform exposure**
Over he course of three weeks. . .

Does exposure to the platform features (Platform Group) affect self-efficacy differently to merely tracking daily prosocial behavior (Tracking-Only Group)?

**RQ4: Improving the platform**

What steps could be taken to improve the platform in the future in order to increase self-efficacy, support suggested action completion and daily prosocial behavior?

## Method

### Study design

The experiment had a longitudinal mixed design (Table 2). The outcome variables of interest were *daily prosocial behavior, suggested actions completed*, and two measures for self-efficacy: *change impact self-efficacy* and *everyday helping self-efficacy*. The between-subject experimental variable was platform exposure with two levels, "Platform Exposure" (Platform Group) and "No Platform Exposure" (Tracking-Only Group). The within-subject variable was time, with daily measurements over the course of 21 days for daily prosocial behavior, and weekly measurements over the course of three weeks for all other variables. Exclusively measured for the Platform Group were platform *enjoyment*, *appreciation* (an additional gratification measure), and *usability* (a control variable to determine the extent to which the platform allowed for an effective, efficient and satisfactory experience). Additionally measured across all participants were *general self-efficacy* (in order to compare with the two principal measures of self-efficacy variables) and *wellbeing* (in order to examine previously found effects that prosocial behavior is related to increases in wellbeing). The quantitative examination of these variables throughout the study was complemented with qualitative post-study interviews for a mixed methods exploration of the research questions.

### Materials

**Interactive platform: Designing for behavior change.** The personalized interactive online platform used was a functioning prototype called *Simple Acts*, which was conceptualized and designed by the first and second author and implemented by the sixth author.

The interactive platform was designed to support the behavior change desired in accordance with the COM-B model for behavior change through intervention [29], and in alignment with literature using behavioral tracking and virtual incentives to prompt changes in behavior [5]. The techniques utilized are summarized in Table 3. Michie et al [29] describe intervention functions as having multiple components that can be facilitated through technology, such as psychological capability being facilitated through providing timely feedback, social opportunities being facilitated through the provision of non-verbal social rewards, and the provision of incentives and rewards (even non-tangible) when undertaking desired behaviors. The Simple Acts platform is therefore an ideal avenue for supporting these behavioral interventions, as digital technology supports immediate and personalised feedback to participants.

**Table 2. Experimental study design.**

| Variable Type | Variables Measured |
|---|---|
| Outcome Variables | Daily prosocial behavior, suggested actions (Platform Group), change impact self-efficacy, everyday helping self-efficacy |
| Between-Subject Variable | Platform Exposure (Platform Exposure, No Platform Exposure) |
| Within-Subject Variable | Time (0-21 days; 0-3 weeks) |
| Covariables & Control Variables | Enjoyment, Appreciation & Usability (Platform Group), General Self-Efficacy, Wellbeing |

**Table 3. Behavior change techniques utilized by the platform.**

| Behavior Change Technique (Michie et al 2014) | Platform components |
|---|---|
| Feedback on behavior | Immediate Feedback: Upon completion of an Action, participants were provided with immediate positive feedback. This has been found to have a significant and substantial impact upon participant behavior [26, 50]. |
| Incentive | Virtual Rewards: Badges provided at the completion of an Action helps participants visualise the impact that they can have on the world around them. Intangible but symbolic incentives such as badges are often found to have positive impacts on the perception of system-based activities, compared to tangible or less subtle extrinsic reward mechanisms [51]. |
| Identity associated with changed behavior | Virtual Rewards: As above. Progress Visualisation: Participants are presented with a map of their location, and are able to decorate it by placing the Virtual Reward icons upon it, thereby tracking their personal progress. The use of personally meaningful representations have been found to contribute towards feelings of identity that can be associated with desired behaviors, thereby increasing the likelihood of long term display of such behaviors and supporting mastery [29]. |
| Information about social and environmental consequences | Personalised Suggested Actions: Participants are presented with information about the societal and environmental benefits that these actions can have, as reasons supporting the display of these behaviors. The provision of contextual information surrounding desired behaviors has been shown to be beneficial to supporting behavior change [5]. |

Towards this goal, the Simple Acts platform was designed specifically to support several Behavior Change Techniques [29] related to feedback, as well as both reflective and automatic motivational affordances commonly found to be beneficial for changing behavior through gamified and persuasive technologies [26, 52]. It was expected that the use of positive feedback mechanisms within the virtual platform would provide a support-based intervention that educates and models prosocial behaviors [53]. For example, the use of overt or extrinsic incentives and rewards has been associated with detrimental long term impacts upon displayed behavior, motivation to undertake behavior, and self-efficacy [54]. In contrast, studies that examined the effect of positive feedback in subtle but personally meaningful ways have been found to better support positive outcomes such as feeling competent and contributing towards social goals [55] and supported longer term engagement with behavioral interventions through the development of mastery [26, 56].

**Personalization of suggested actions.** While the platform was fully functional, it worked in a relatively simple fashion. Personalization was achieved by collecting information on platform users' interests (to personalize to motivation) and activities (to personalize to capability) prior to granting access. For this study, our users were all students in the same university town, making personalization to opportunity easier as we could focus on opportunities in one location. Table 4 summarizes how personalization to the COM-B model was achieved.

**Table 4. Platform personalization based on the COM-B model.**

| Behavioral Facilitator | Personalization based on |
|---|---|
| Capabilities | Current activities participants listed as enjoying (e.g., making others feel appreciated, going outside, getting to know new people). |
| Opportunities | Location (all participants were students in the same university town). Brevity of actions. |
| Motivation | Interest in topics listed by participants as being important to them (e.g., supporting local business, animals, poverty, disability) |

To personalize suggested actions to their capabilities, opportunities, and motivations, participants were asked to rate twelve interests and eight activities on a scale from 1 (not at all interested) to 5 (very interested). An action database was created by the fourth and fifth author based on prosocial activities, which could be undertaken in and around the university town, supporting their opportunities. These activities could be performed without long-term commitments and in a relatively short amount of time, again supporting opportunity, varying between a few seconds (saying "thank you" to the bus driver) and a few hours (playing football with a group, striving to connect locals and foreigners in the local community). The action database consisted of 55 actions, which were dichotomously coded (0 or 1) for whether they related to each activity and interest. Based on participants' answers to the activities and interests questions, they could therefore be assigned a fit score for each action, indicating to which extent these actions matched their motivations and capabilities. Over the course of the three weeks participants in the Platform Group were assigned the nine actions (three per week) with their individually highest fit scores. Participants could then choose which (if any) of these actions they wanted to complete. In order to ensure that the completion of actions was not inhibited by the financial capacity of participants, each participant's suggested actions selection was checked to ensure that each participant received no more than one donation-related action per week.

**Interactive platform: Participant interactions.** Using their unique participant code, participants in the Platform Group could log into the platform over a browser of their choice on either desktop or mobile devices. After logging in, participants saw an interface that displayed a map of a university town (see Fig 1) and three suggested actions (see Fig 2). These suggested actions were prosocial activities, which had been personalized to the participant based on their interests in issues and preferred activities (see section "Personalization of suggested actions" for more detail on the personalization). Each suggested action was described with a colorful icon, a title, a one-line subtitle, a paragraph on the reason why this activity was important, and a step-by-step guide of how to complete the action (see Fig 2 for an example). Additionally, each suggested action description included a button, with which participants could report having completed that action. The button took participants to a short survey, in which they could indicate how easy the suggested action had been to complete and whether they had any issues while completing it. Once an action was marked as completed, participants received a virtual reward in the form of a badge (the icon for that action), which, to visualize their progress, on desktop could be dragged-and-dropped onto the map on the approximate location at which the action had been completed.

## Measurements

In the following, the questionnaires, items, and interview questions used in this study are presented. All communication and measures were displayed to participants in German, the official language in the university town where this study was conducted. In this chapter and throughout the rest of the manuscript items and quotes are English translations. The German original material can be found on the Open Science Framework (OSF), where this project can be found under the following link: https://osf.io/dnu3y/.

**Daily prosocial behavior.** Daily prosocial behavior was measured using the question "Please think back on yesterday. How many actions did you do that you would describe as prosocial?". Additionally, they were asked to list all actions that they could think of in an open text field. To ensure that participants across the study were thinking about prosocial behavior in a comparable way, all participants were given a definition of prosocial behavior. This definition described prosocial behavior as voluntary, not related to material gain, and defined by a

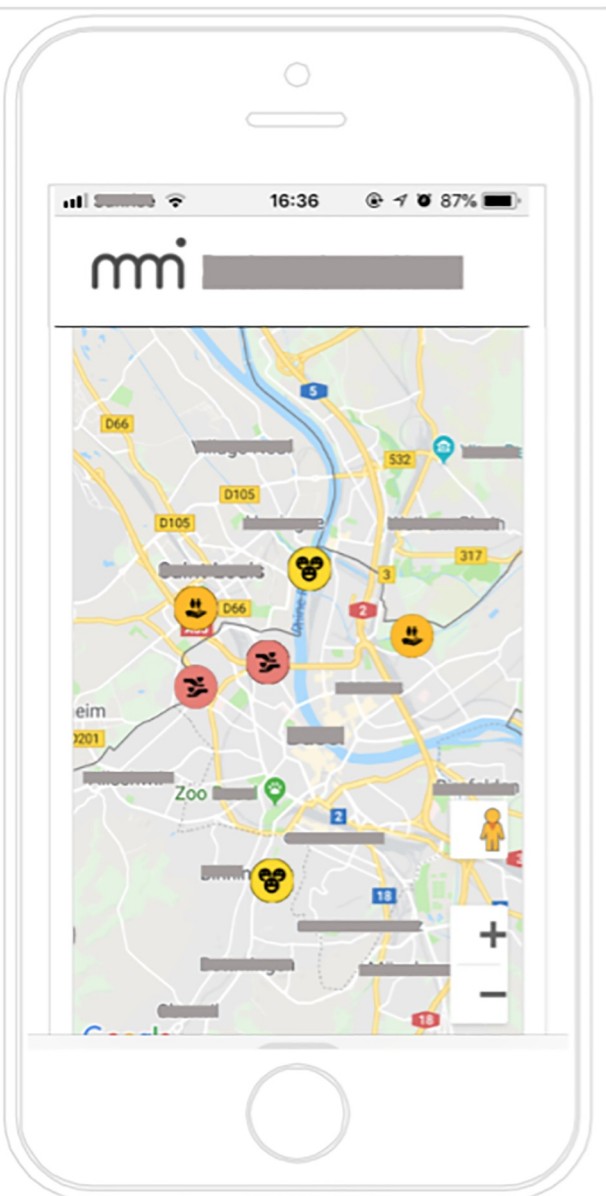

**Fig 1. The map (progress visualization) with badges (virtual rewards).** Participants received badges for actions they had completed. They could pull the badge onto the place on the map where they had performed the action to visualize their progress. Map location and research group name masked for review.

positive effect on others. They were also given examples of prosocial behavior, including giving up one's seat in public transport, donating, or letting someone know that they are appreciated.

Daily prosocial behavior scores were the mean number of daily actions a participant reported. For week-level analysis, these means were averaged across the week.

**Suggested actions completed.** Suggested actions completed was tracked using the platform, on which participants could mark actions as completed. Participants could complete between zero and three actions per week.

**Self-efficacy.** As previously discussed, self-efficacy is most accurate as a predictor of behavior, when it is measured as specifically as possible. Therefore, we focused on two

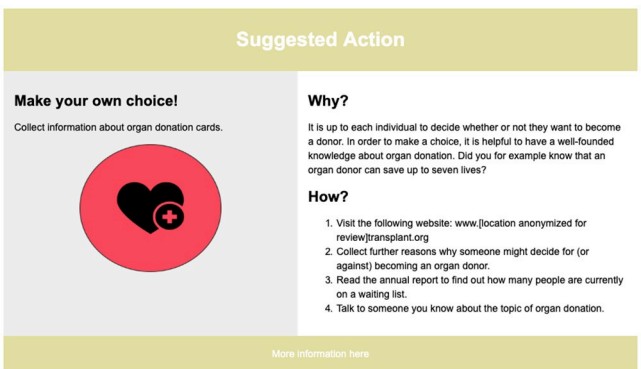

**Fig 2. Example of a suggested action.** This suggested action, named "Make your own choice!", explained to participants how they could find out more about organ donation and why it was important to make an informed choice about whether or not to carry an organ donor card. Under the Why?-section, participants were told that signing up to become an organ donor after death could lead to up to seven lives being saved. Under the How?-section, participants were given a step-by-step guide on how to find more information about organ donation, including exploring the official national organ transplant page, making a pro/con list for themselves, and talking to someone about the topic.

particularly relevant forms of self-efficacy. Firstly, change impact self-efficacy, building on the previous conceptualization by White et al. [17]. Secondly, everyday helping self-efficacy, the belief in one's ability to perform, even in the face of challenges, everyday actions, which benefit others. As no measurement for everyday helping self-efficacy existed, a scale was developed in two pilot studies and a preliminary validation performed during the main study. A measure for general self-efficacy was also used for inclusion in correlational analyses.

**Change impact self-efficacy**. To measure change impact self-efficacy, the scale by White et al. [17], which was developed in the context of fair trade purchasing choices, was modified to the more general context of this study. The scale consisted of four items (*"I believe that small actions can make a difference", 'I believe that the actions, which I undertook this past week can improve the lives of others'*, *"I believe that I can positively impact the lives of others.", "I believe that the actions, which I undertook this past week can improve society at large."*). Each statement could be evaluated on a scale from 0 (Not at all) to 100 (Completely). The scale had high internal consistency (Cronbach's $\alpha$ = .79).

**Everyday helping self-efficacy.** To our knowledge no measures exist, which have been built to examine self-efficacy specifically in the context of everyday prosocial behavior. Therefore, in order to measure everyday helping self-efficacy, prior to the longitudinal study, we conducted two pilot studies in order to develop a scale to measure everyday helping self-efficacy.

*Pilot study 1.* We began by conducting a qualitative online survey in which 28 participants answered the questions "What makes it difficult for you to act prosocially in an everyday context" and "What makes it easy for you to act prosocially in an everyday context?" [57]. The responses were analyzed and discussed among three of the authors. Based on this discussion, similar responses were combined and responses with more than one statement were separated. The resulting list of statements were re-written into 46 items, which began with the prephrasing "To what extent do you believe in your abilities to successfully help in an everyday situation, if. . .". These items could be answered on a scale from 0 (Not at all) to 100 (Completely) [57].

*Pilot study 2.* The 46 items were given to a new sample of 62 participants. The participants responded to all prepared items and additionally rated a subsample of 13 to 26 items in terms

of how understandable they were.Understandability was measured on a 5-point Likert scale from 1 (Not at all) to 5 (Completely).

The items were then analyzed in terms of their understandability, difficulty indices, discriminatory power, and inter-item correlation (values can be found on the Open Science Framework). Based on these analyses, 14 items were removed and one item was rephrased to be better understandable.

Based on the data from the baseline measurement of the longitudinal study, these 32 items were evaluated using a principal component analysis, based on the procedure described by Field et al. [58]. An initial analysis was run to obtain eigenvalues for each component in the data. Three components had eigenvalues over Kaiser's criterion of 1. However, due to the preliminary nature of this questionnaire it was decided to maintain a single component structure and focus on a smaller number of items similar to that of other questionnaires used in this study (between 3 and 6 items). Items with commonalities lower than.5 on the primary component were removed (26 items). The final analysis was conducted with the remaining six items (Table 5). Factor loadings for all six items were above.7. The six items were evaluated on a scale from 0 (Not at all) to 100 (Completely). The scale correlated moderately with general self-efficacy (r = .47, p <.001), highly with change impact self-efficacy (r = .53, p <.001) and had a high internal consistency (Cronbach's $\alpha$ = .90).

**General self-efficacy.** To measure general self-efficacy, the scale developed by Chen et al. [14] was utilized. The six items (e.g., *"I believe I can succeed at most any endeavor to which I set my mind."*) were measured on a 7-point Likert scale from 1 (strongly disagree) to 7 (strongly agree). The scale had high internal consistency (Cronbach's $\alpha$ = .93).

**Platform experience.** Behavioral facilitators have, amongst other things, the purpose of making the interaction with a platform more enjoyable and thereby encourage increased and sustained interaction. In line with this, past research has shown that technology-based behavioral facilitators can lead to increased enjoyment when engaging with tasks related to tasks such as learning [48] and exercise [26] Furthermore enjoyment has been used as an effectiveness measure for digital game-based learning [59]. Interactivity and prosocial behavior were previously shown to be related to appreciation [41]. A similar connection was therefore expected in this study. Since media can be both appreciated and enjoyed at the same time [48] both concepts were measured. To ensure the continued usage of a platform or system it has been shown that user experience is a crucial element [46–48]. Since the present study required participants to use the gamified platform for ideally two weeks, it was important to create and ensure a usable and enjoyable platform.

**Table 5. The six items of the *Everyday Helping Self-Efficacy Scale*.** These items were generated in pilot study 1.

***To what extent do you believe in your abilities***
***to successfully help in an everyday situation, when*...**

| Item | Factor Loadings |
|---|---|
| ...you are feeling angry. | .82 |
| ...it feels as if your help is unimportant | .81 |
| ...no one will thank you for helping. | .79 |
| ...you are afraid you might do something incorrectly. | .77 |
| ...you see that no one else is helping. | .76 |
| ...in the past you failed at a similar attempt to help. | .73 |
| Eigenvalue | 3.64 |
| % of variance | 60.73 |

**Enjoyment.** In order to measure enjoyment, the audience response scale for enjoyment, developed by Oliver and Bartsch [49] was adapted to the study context. The three items (e.g., *"Working with the Simple Acts platform was fun for me"*) had a high internal consistency (Cronbach's $\alpha$ = .83).

**Appreciation.** In order to measure appreciation, the audience response scale for appreciation, developed by Oliver and Bartsch [49] was adapted to the study context. The three items (e.g. *"Working with the Simple Acts platform was very meaningful for me"*) had a high internal consistency (Cronbach's $\alpha$ = .92).

**Usability.** Usability was measured using the Usability Metric for User Experience (UMUX) scale developed by Finstad [60]. The scale consists of four items, one each to measure efficiency (*"I have to spend too much time correcting things with this platform"*; reverse scored), effectiveness (*"This platform's capabilities meet my requirements."*), satisfaction (*"Using this platform is a frustrating experience"*; reverse scored), and an overall usability item (*"This platform is easy to use."*). Items are evaluated on a 7-point Likert scale from 1 (Strongly disagree) to 7 (Strongly agree). Item scores subtract 1 (or are subtracted from 7 for reverse scored items). The UMUX score is the sum of the four items, divided by 24 and multiplied by 100, which leads to a range of 0 to 100 (Cronbach's $\alpha$ = .64; see Finstad, [60] for a discussion of the UMUX score calculation).

**Wellbeing.** In order to measure wellbeing, the 5-item World Health Organization Well-Being Index (WHO-5) was used, which has been repeatedly shown to consist of simple and non-invasive questions, which can measure wellbeing across a range of study fields (see Topp et al. [61] for an overview). The items were adapted to the study context to refer to a one-week period, instead of a two week period. Items (e.g., *"Over the past week, I have felt active and vigorous"*) were scored on a 5-point Likert scale from 0 (At no time) to 5 (All the time). The scale had high internal consistency (Cronbach's $\alpha$ = .87).

**Likelihood to continue.** In order to understand whether participants would wish to continue using the platform in the future, a single item at the end of the study asked how likely participants would be to use the platform in the following week, if the platform were openly available. The item was measured on a 7-point Likert scale from 1 (Very unlikely) to 7 (Very likely).

**Interview questions.** Following the study, a post-study interview was conducted in order to understand what worked well and what could be improved upon for future iterations of the platform. The interview was conceptualized and run as semi-structured, allowing the interviewer to adapt questions to each participant while still following a general structure across participants.

Participants were asked a series of questions related to their experiences with the behavior tracking and how tracking had impacted the way they thought about their prosocial behavior on a day-to-day basis (e.g *"How did you experience being confronted with questions around prosocial behavior on a daily basis?"*). Additionally, participants in the Platform Group were asked about their experiences interacting with the platform and how it could be improved upon, especially in order to support prosocial behavior more effectively (e.g., *"What would you like to see changed about the platform? What would you definitely want to keep?"*). Questions were asked both about the platform in general as well as about specific features, such as the personalized suggested actions and the map.

## Participants

68 participants began the longitudinal study. Two participants were excluded for filling out only one entry in the weekly questionnaires. 18 participants had dropped out by the third

weekly measurement (eight from the Platform Group). As multilevel analyses were utilized to examine the research questions, existing data for participants with missing data could be incorporated into the analysis [58]. Therefore, data from 66 participants (33 in each condition) were included in the study. 78 percent of participants identified as female. Participants ranged in age from 18 to 37, with a mean age of 22.7 years (SD = 3.2). All participants were bachelor students of psychology. Participants were recruited over the faculty study advertisement platform. The advertisement explicitly mentioned prosocial behavior and its facilitation as a central part of the study. Participants were rewarded with course credit for each weekly survey they participated in; The incentives were dependent neither on the amount of platform usage, nor daily prosocial behavior reported. Participants were allocated to conditions randomly. Research was conducted in accordance with university guidelines on ethical practices for Psychology research. Written informed consent of the participants was obtained before beginning the study. Participant data was anonymized.

## Procedure

The study measured data at six different timepoints: Prestudy, baseline, week 1, week 2, week 3, and post-study interviews. Which variables were measured at which time points is summarized in the following sections. The procedure is additionally illustrated in Fig 3.

**Pre-study.** After signing up for the study, participants filled out a 15-minute pre-study questionnaire, in which they answered questions used to personalize suggested actions based on their preferences regarding activities and interests. Subsequently, participants received a personalized participant code and answered basic demographic questions (age, gender, occupation).

**Baseline measurement & week 1.** The following week, participants began the three-week study, which began with a baseline measurement for both 2-minute daily (daily prosocial behavior) and 15-minute weekly measurements (change impact, everyday helping, and general self-efficacy, as well as wellbeing). For the next 21 days all participants then tracked their daily prosocial behavior by responding to a daily email reminder. Weekly reminders prompted participants to fill out the weekly questionnaires. Additionally, the Platform Group were given access to the Simple Acts platform after the baseline measurements. They were instructed on how to log in but allowed to choose if and how they used the platform. Each week, the platform would suggest three new actions, which participants could then choose to complete if they

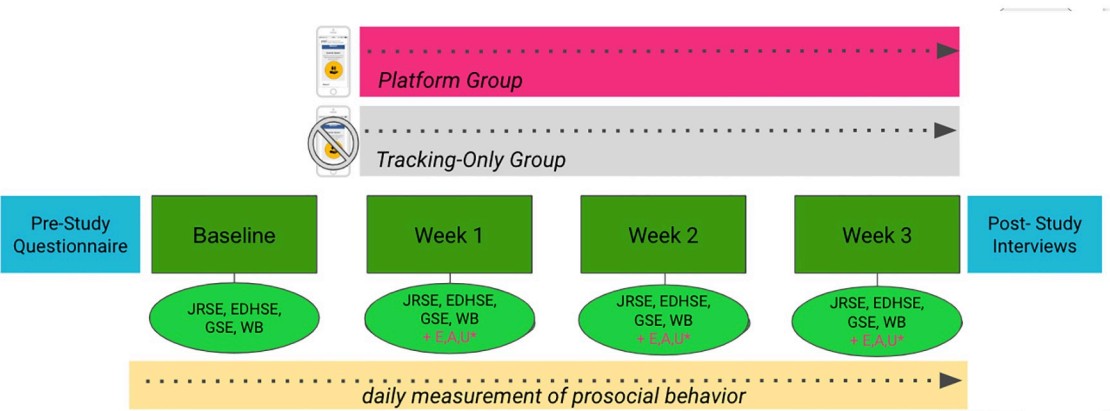

**Fig 3. Study procedure.** An overview of the three week study procedure, including the experimental manipulation, the pre-study questionnaire, baseline, week 1, 2, and 3 measurements, and the post-study interviews.

wished. After the first week, for participants in the Platform Group, the weekly questionnaire included additional questions on the platform experience. These questions were on enjoyment, appreciation, and usability of the platform.

**Week 2 and 3.** At the end of week 2 and week 3, participants answered the same set of questions as after week 1. In addition, at the end of Week 3, participants from both groups received an additional question asking whether they would be interested in participating in a final post-study interview. Participants in the Platform Group were additionally asked how likely they would be to continue using the platform in the next week, if it were openly available.

**Post-study interviews.** Eight participants were included in the exploratory post-study interview, which was conducted in person at the university research facilities. As the focus of the interview was on the platform and how to improve it, six of eight participants were selected from the Platform Group. The number of participants chosen was based on the recommendations given by Clarke et. al. [62] for a research project of small to medium size. Preliminary screening of the interview data showed medium to high saturation of all major topics (see [63, 64]) was reached fairly quickly, at around 5 to 6 interviews, depending on topic. This is also helped by the fact, that the group interviewed was fairly homogeneous and the interviews followed a semi-structured approach, which generally makes it easier to uncover the major topics in a sample [65].

The participants were asked questions relating to their experiences and the ways in which these experiences and the effectiveness of the platform to promote prosocial behavior could be improved.

## Results

All statistical analysis was performed in R [66] using an alpha level of.05. The data sets and R scripts used in the analysis can be found on the OSF. To analyze effects over time, multilevel modeling [67] was used following the procedure described by Field et al. [58]. To examine qualitative interview data, reflexive thematic analysis was conducted as outlined by Braun and Clarke [68] and Clarke et al. [62].

### Data preparation

Data from five different datasets were taken into account for the following analyses. Firstly, the dataset from the pre-study questionnaire containing demographic information on the participants. Secondly, the dataset from the main study containing the measurements of daily prosocial behavior. Thirdly, the dataset from the main study containing the measurements of weekly change impact self-efficacy, everyday helping self-efficacy, enjoyment, appreciation, usability, general self-efficacy, and wellbeing. Fourthly, the dataset from the main study containing each Platform Group participant's completed suggested actions. Fifthly, the dataset containing the transcripts of the post-study interviews. All datasets but the fifth were linked based on the participants' designated codenames. The fifth dataset, which pertained to the in-person interviews was not linked to the rest of the data to maintain participant anonymity throughout the study. The second, third, and fourth dataset were merged for analysis. After the merger, variable distributions were examined and four outlier values removed from daily prosocial behavior.

### Descriptive and correlative results

Means, standard deviations, and measurement ranges for the Platform Group and the Tracking-Only Group at the end of week 3 can be seen in Table 6. Results show that both groups performed on average between one and two prosocial actions per day in the third week. In addition, participants in the Platform Group had performed on average between one and two

**Table 6. Descriptive statistics by condition.**

| | Tracking-Only Group | Platform Group |
|---|---|---|
| **Variables**<br>*(Range)* | **M (SD)** | **M (SD)** |
| Daily prosocial behavior<br>*(0-5)* | 1.32 (0.97) | 1.29(.97) |
| Total completed suggested actions<br>*(0-6)* | - | 1.28 (1.42) |
| Change impact self-efficacy<br>*(33.30-95.50)* | 69.21 (13.49) | 67.90 (17.39) |
| Everyday helping self-efficacy<br>*(16.83-93.33)* | 58.38 (23.54) | 53.57 (19.99) |
| Enjoyment of platform<br>*(1.00-6.67)* | - | 4.21 (1.54) |
| Appreciation of platform<br>*(1.00-6.67)* | - | 3.69 (1.64) |
| Usability of platform<br>*(45.83-100.00)* | - | 70.66 (17.99) |
| Likelihood to Continue<br>*(1-7)* | - | 3.21 (2.00) |
| General self-efficacy<br>*(2.33-7.00)* | 5.69 (0.86) | 5.60 (1.04) |
| Wellbeing<br>*(1.00-5.00)* | 3.29 (0.89) | 2.87 (0.94) |

of the possible nine suggested actions by the end of the study. Results also show moderate values for change impact self-efficacy and everyday helping self-efficacy, as well as enjoyment, appreciation, usability, and likelihood to continue using the platform.

Pearson correlations across groups can be seen in Table 7. As can be discerned here, all self-efficacy scales correlate moderately to highly with one another, as do enjoyment, appreciation, and usability. Daily prosocial behavior correlates most strongly with change impact self-efficacy, while completed suggested actions correlates most strongly with enjoyment of the platform. As can be seen in Table 7, higher enjoyment, appreciation, and usability ratings were all associated with a higher likelihood of continuing use of the platform.

### Exploring research questions 1-3 with multilevel models

In order to examine RQs 1-3, six multilevel models were analysed. These models took into account both within-subject effects over time, as well as between-subject effects due to platform exposure and random variance in intercepts and slopes. In preparation, variables were grand mean centered [58]. Model parameters can be found in Table 8.

**RQ1: Daily prosocial behavior & platform usage.** As can be seen in model RQ1a (Table 8), platform exposure did not significantly predict daily prosocial behavior ($b = .08$, $p = .772$). Time was significantly negatively associated with daily prosocial behavior ($b = -.31$, $p < .001$).

In model RQ1b (Table 8), change impact self-efficacy additionally positively predicted daily prosocial behavior ($b = .02$, $p = .012$). Everyday helping self-efficacy did not significantly predict daily prosocial behavior ($b = .001$, $p = .865$).

This means that platform exposure did not impact daily prosocial behavior more than simply tracking daily prosocial behavior. Change impact self-efficacy positively predicted daily

**Table 7. Pearson correlations.**

|  | Completed suggested actions | Change impact self-efficacy | Everyday helping self-efficacy | Enjoyment of platform | Appreciation of platform | Usability of platform | Likelihood to Continue | General self-efficacy | Wellbeing | Time |
|---|---|---|---|---|---|---|---|---|---|---|
| Daily prosocial behavior | .01 | **.29***** | **.15*** | .15 | .20 | .02 | -.20 | .12 | **.17*** | **-.20**** |
| Completed suggested actions | | **16*** | .02 | **.25*** | .18 | .12 | .05 | .06 | -.004 | .06 |
| Change impact self-efficacy | | | **.53***** | **.39***** | **.33**** | **.33**** | .19 | **.39***** | **.38***** | -.01 |
| Everyday helping self-efficacy | | | | **.49***** | **.37***** | **.38***** | .39 | **.47***** | **.42**** | **.16*** |
| Enjoyment of platform | | | | | **.78***** | **.49***** | **.55**** | **.26*** | **.45***** | -.14 |
| Appreciation of platform | | | | | | **.30**** | **.66***** | .14 | **.22*** | -.10 |
| Usability of platform | | | | | | | **.51*** | **.40***** | **.27***** | -.05 |
| Likelihood to Continue | | | | | | | | **.42*** | .15 | - |
| General self-efficacy | | | | | | | | | **.47***** | .10 |
| Wellbeing | | | | | | | | | | **.13*** |

* $p < .05$.

** $p < .01$.

*** $p < .001$.

prosocial behavior, while everyday helping self-efficacy did not. Reports of daily prosocial behavior decreased over the course of the three weeks.

**RQ2: Suggested action completion using the platform.** In model RQ2a (Table 8), we examined the relationship between self-efficacy and the number of weekly suggested actions participants in the Platform Group completed. Change impact self-efficacy significantly predicted completion of suggested actions ($b = .01$, $p = .017$), but everyday helping self-efficacy did not ($b = -.01$, $p = .271$). Time was significantly negatively associated with completion of suggested actions ($b = -.19$, $p = .035$).

In model RQ2b (Table 8), enjoyment of the platform predicted completion of suggested actions ($b = .12$, $p = .034$), while time did not ($b = -.17$, $p = .058$).

This means that completion of suggested actions was predicted by change impact self-efficacy and enjoyment of the platform, but not by everyday helping self-efficacy. Participants completed less suggested actions as time progressed, although the inclusion of enjoyment in the model lead to this relationship no longer being significant.

**A note on multicollinearity.** To account for the potential effect of multicollinearity, the effects of change impact self-efficacy and everyday helping self-efficacy were isolated in separate models for RQ1b and RQ2a. For RQ1b, the two models showed effects in the same direction as the combined RQ1b model. That is, a significant effect for change impact self-efficacy ($b = 0.01$, $p = 0.005$) and no significant effect of everyday helping self-efficacy ($b = 0.01$, $p = 0.2822$). For RQ2a, the two models as well showed effects in the same direction as the combined RQ2a model. That is, a significant effect for change impact self-efficacy ($b = 0.01$,

**Table 8. Multilevel models for research questions 1-3.**

| Predictor Variables | Model RQ1a Outcome Variable: Daily prosocial behavior | | Model RQ1b Outcome Variable: Daily prosocial behavior | | Model RQ2a Outcome Variable: Suggested actions completed | | Model RQ2b Outcome Variable: Suggested actions completed | | Model RQ3i Outcome Variable: Change impact self-efficacy | | Model R 3ii Outcome Variable: Everyday helping self-efficacy | |
|---|---|---|---|---|---|---|---|---|---|---|---|---|
| | *b* | *SE* | *b* | *SE* | *b* | *SE* | *b* | *SE* | *b* | *SE* | *b* | *SE* |
| **Fixed Effects** | | | | | | | | | | | | |
| Time | **-.31**\*\* | .08 | **-.30**\*\*\* | .08 | **-.19**\* | .09 | -.17 | .09 | -.57 | .83 | **2.86**\*\* | .80 |
| Platform Exposure | .08 | .28 | .03 | .27 | | | | | 2.94 | 3.50 | -1.03 | 4.37 |
| Change impact self-efficacy | | | **.02**\* | .01 | **.01**\* | .01 | | | | | | |
| Everyday helping self-efficacy | | | .001 | .01 | -.01 | .01 | | | | | | |
| Enjoyment | | | | | | | **.12**\* | .06 | | | | |
| **Random Effects (participant level)** | | | | | | | | | | | | |
| Random intercept variance | **1.35**\* | | **1.35**\* | | 0.06 | | **0.07**\* | | 218.02\* | | 122.94\* | |
| Random slope variance | **.11**\* | | **0.08**\* | | .0002 | | .0001 | | 22.49\* | | 5.59\* | |
| Random intercept / Random slope correlation | -.98 | | -.98 | | -.11 | | .01 | | **-.46**\* | | .34 | |
| AIC/BIC | *716.49/743.96* | | *713.27/747.61* | | *193.78/215.87* | | *191.43/208.61* | | *1928.50/1952.80* | | *1912.86/1940.68* | |
| Fixed Effects R² | **.08**\* | | **.13**\* | | **.12**\* | | **.11**\* | | **.01**\* | | **.06**\* | |

\* *p* <.05.

\*\* *p* <.01.

\*\*\* *p* <.001.

*p* = .029) and no significant effect of everyday helping self-efficacy (*b* = 0.002, *p* = 0.639). This suggests that multicollinearity did not bias the results.

**RQ3: Self-efficacy & platform exposure.** In model RQ3i (Table 8), neither time (*b* = -.57, *p* = .49) nor platform exposure (*b* = 2.94, *p* = .304) were associated with increases in change impact self-efficacy. In model RQ3ii (Table 8), time was associated with increases in everyday helping self-efficacy (*b* = 2.86, *p* <.001), while platform exposure was not (*b* = -1.03, *p* = .815).

This means that over the course of the three weeks, participants gained everyday helping self-efficacy, but not change impact self-efficacy. Platform exposure affected neither form of self-efficacy.

**Examining wellbeing over time.** As the correlational analysis had shown a positive relationship between wellbeing and time, we decided to examine wellbeing over time and across groups with a further multilevel model (Random intercept variance = -0.01, *p* <. 05; Random slope variance = 0.01 *p* <. 05, Random intercept-random slope correlation = -.20, *p* >.05; AIC = 567.07, BIC = 591.40, Fixed Effects R² = .03, *p* <.05).

Results found time to significantly positively predict wellbeing (*b* = .11, *p* = .016), while platform exposure did not (*b* = -.09, *p* = .63).

## Exploring research question 4: Improving the interactive platform

In order to examine RQ 4, the data from the post-study interviews were explored using thematic analysis, a technique for the analysis of qualitative research developed by Braun and Clarke [68]. Thematic analysis utilizes a systematic and rigorous approach to coding and theme development and purposefully avoids the quantification of qualitative results, emphasizing the researcher's active role in the research process (see Clarke et al. [62] for an excellent overview on thematic analysis).

**Coding procedure.** Interview transcripts were used to perform a reflexive thematic analysis of the interview material [62, 68]. An open, data driven, inductive approach was used. The interviews were conducted by the third and fourth author, then transcribed and analyzed by the third author in continuous discussion with the first author. As suggested by Braun and Clarke [68], analysis commenced with data familiarization and open coding. The codes were also sorted into topics, checked for saturation and were then developed into themes in further iterations. To keep track of the coded passages, the qualitative data analysis program MaxQDA was used.

The analysis resulted in five main themes: (1) *Being aware of situations where one can help and one's attitude towards helping,* (2) *Fun and Empowering* (3) *Appreciated but easily forgotten* (4) *The importance of fitting seamlessly into daily life* (5) *The need to see an impact.*

**Being more aware of situations where one can help and one's own attitude towards helping.** While participants reported that daily tracking of the activities was a bit boring, they also reported that tracking their prosocial behavior made them think about prosocial behavior more and that they became more capable of spotting situations in which they could act prosocially. This was mainly attributed to the fact that they were confronted with the themes of prosocial behavior and helping others daily. This is showcased by the following excerpts.

"To me it feels like I got more perceptive, that I saw more situations where I actually could help."

(P6)

"The study generally got you thinking about the topic [of prosocial behavior]."

(P4)

Next to being more aware of situations in which they could help, participants reported that being confronted with prosocial behavior daily made them more aware of their own attitude towards the topic of.

"[The study] was exciting. You could see what prosocial behavior you are really showing and if you could do more. You questioned your behavior more and more each day."

(P8)

These reports, however, were tempered by participant doubts that any changes would last beyond the run of the study. While participants felt that their perception of and their attitude towards prosocial behavior had changed, they did not expect lasting changes in their behavior. They felt that they had changed their behavior throughout the three week study, but they were not sure how they would continue after the study when they were no longer confronted with the topic daily.

"The perception changed more than the actual behavior. Maybe the behavior changed more when it came to little things, like inviting someone to dinner. But 'bigger' prosocial behavior was harder to show. [. . .] It's just that with these things, it needs more time."

(P9)

**Fun and empowering.** Participants in the Platform Group reported generally enjoying the platform.

"[The platform] was very appealing, with its weekly actions, bringing books to a thrift shop in [town], clothing as well."

(P2)

Beyond this, participants who completed suggested actions reported feeling empowered by the guidance they received on the platform.

"I felt more responsible, I also helped [when I was not directly affected]. For example, I walked by a trash can and the trash was lying beside it. First I was like, why didn't people do anything about this, but then I thought I can do that myself and put the trash into the can."

(P1)

"I [was on vacation] and picked up stuff at the beach. It is a really nice thought, one gets more aware of what can be done to help."

(P3)

"Even just the feeling [changed]. Just the small things. When I cook, then I'm helping the people around me. I won this belief that I am capable of helping others."

(P2)

**Appreciated, but easily forgotten.** While the general concept of the platform seemed to appeal to the participants, for the most part they engaged only irregularly with it. Reasons therefore seemed to lie in the fact that the platform was not particularly easy to use—having to actively be sought out via browser on mobile or desktop. This lead to participants completely forgetting about the platform for most of the time.

"Often times I looked at the actions at the beginning of the week and checked out which ones I could do and thought, ohh this one and that one, but then the week was suddenly over."

(P1)

To counteract this, participants suggested that the platform could have regularly reminded them of the actions they had been assigned because they forgot about them during the week.

"A reminder would have been nice, maybe as a push message on my mobile."

(P1)

"I think it would be really cool if the platform would remind you via push or email."

(P10)

Another issue could have been that features designed to reward and encourage participants were not perceived as interesting or useful enough to support across different situations and over time. For example, while the map was considered to be a nice detail, it was not seen as particularly helpful or important.

"The problem with the map was that I live in [another town] and the map was centered on [the university town]. I was not sure if or how I could have changed that."

(P10)

**The importance of fitting seamlessly into daily life.** Overall the personalization of actions was either something that participants were not aware of until told during the interview or something that was considered helpful.

"[The personalization] worked well, I got offered actions that I could easily take care of."

(P2)

However, several issues participants reported revolved around a lack of fit between the platform, its actions, and their daily life. For one, because performing the suggested actions often necessitated going to specific places, participants wanted to use the platform across different types of devices. In general however, mobile was the preferred way to access the platform.

"'I used the platform with the mobile, it worked pretty well. I think the mobile is very important, so that you can do the actions on-the-go."

(P2)

While the platform was available to use on mobile, this was over a mobile website. Different participants suggested that the platform could be run as a mobile application. This could either be a newly created app or an integrated feature of apps they already use.

"An app would be nice, but it could be integrated into an app that already gets used, for instance email, social media like Instagram or Facebook."

(P1)

Participants expressed the need for suggested actions to fit into their daily schedule. They criticised the fact that they had at times been offered suggested actions that did not fit into their time budget or current situation.

"[When choosing an action], my stress level as well as the needed time to complete it are important. I would say the time it takes to complete an action is the most important [thing I consider]."

(P6)

Relatedly, participants criticized the limited selection of suggested actions. During the study, participants received three suggested actions per week. In the interviews the wish was expressed to have been offered more options, which would have allowed them more flexibility to find something fitting their current circumstances.

"It would be cool if there was a bigger selection of actions to choose from. (. . .) Filtering [the actions] would be great, especially by effort, mood, and stress level."

(P6)

"If you had more choices when you're offered actions, maybe 2 or 3 per category, then the chance that (one of those) actions fits is higher."

(P1)

This makes apparent that action personalization needed to understand capability, opportunity, and motivation in more nuanced terms, as they could change based on context, such as their current mood or time restraints. Beyond this, there seemed to be a need for choice.

**The need to see an impact.** An additional issue surfaced, which the selection of suggested actions had not adequately addressed. In advance of the study, care had been given during the action suggestion process to include a maximum of one donation-related action per participant per week. Despite these efforts, the interviews made clear that for some participants calls for donations were simply not interesting and that they preferred actions for which they could directly see the impact.

"I think the personalization worked okay. But let's say you ticked helping animals on the survey, I thought to myself: "yes great". But then I was asked to donate to some animal rights group. . ."

(P3)

"The only thing that matters to me is that whenever I do something that I really help someone. It needs to be meaningful."

(P1)

Above all, actions needed to seem meaningful, with the benefit as apparent as possible. Donations in particular did not seem to allow for enough of a sense of meaningfulness.

## Discussion

The aim of this study was to investigate the impact of a personalized platform on daily prosocial behavior (RQ1), platform-prompted personalized suggested action completion (RQ2), and self-efficacy (RQ3). Additionally, we wished to gain insights into potential approaches to improving the platform in the future (RQ4).

In order to explore these research questions, a three week experiment was conducted, along with a pre-study questionnaire and a post-study interview. Multilevel models were used to explore the study's first three research questions. The post-study interview focused on the fourth and final research question and was examined using reflexive thematic analysis.

Results showed that platform exposure with its features of suggested actions, progress visualization, and virtual rewards impacted neither daily prosocial behavior, nor self-efficacy more than merely tracking daily prosocial behavior. This highlights the importance of understanding how this or other similar platforms can be improved upon in the future in order to be more effective. Post-study interviews were used to understand what had worked and what had not in the current platform version. In these interviews, participants reported feeling more responsible, more aware of situations in which they could help, and more likely to help. They however also described the platform being forgotten about in the complexities of everyday life, as well as wanting more emphasis on the impact they would have with their actions. In the quantitative results we observed that likelihood to continue using the platform was related to higher levels of enjoyment, appreciation and usability of the platform.

Beyond this, higher enjoyment predicted a larger number of completed suggested actions. In order to support engagement over a longer time period these results point to the importance

of ensuring the maintenance of a positive experience over time. The role of enjoyment and usability here is in line with past research on sustained usage [46–48]. The role of appreciation is in line with past research on the positive relationship between appreciation and prosocial behavior [41, 69]; This study, however, is the first time this effect was examined for behavior over time.

Both measured forms of prosocial behavior, completed suggested actions and daily prosocial behavior, were predicted by change impact self-efficacy. This is in line with self-efficacy theory [10]. Interestingly, however, everyday helping self-efficacy did not predict either form of prosocial behavior. Existing research on prosocial behavior repeatedly highlights the importance of knowing that one's actions will make a difference for people to act prosocially [17, 19]. While everyday helping self-efficacy focused on the specific actions involved in daily prosocial behavior, it appears as though this belief was not enough to galvanize participants into prosocial action. Technology wishing to support prosocial behavior change would therefore do well to ensure that users are strengthened, particularly in their belief to affect change.

Both multilevel modeling as well as correlational analysis pointed to everyday helping self-efficacy and wellbeing, but not change impact self-efficacy, increasing over time in both groups. Keeping in mind that causality remains unclear, it is possible that tracking daily prosocial behavior may have led to this increase in everyday helping self-efficacy and wellbeing. This is in line with recently published findings that recalling prosocial behavior can increase wellbeing [38]. As tracking daily prosocial behavior would have made participants aware of the number of prosocial actions they were already doing, it is reasonable to assume that this perception of mastery experiences could also lead to increases in everyday helping self-efficacy. Particularly, as everyday helping self-efficacy, unlike change impact self-efficacy, was focused on the belief in the ability to perform everyday prosocial actions such as the ones tracked by participants. The platform features of personalized suggested actions, progress visualization and virtual rewards were designed with the intention of increasing both everyday helping self-efficacy and change impact self-efficacy. However, as previously discussed, neither form of self-efficacy was increased by platform exposure.

Interview results offer potential avenues in which the platform could be improved upon in the future to better support both forms of self-efficacy as well as prosocial behavior. In the interviews, participants emphasized the importance of seeing the impact they were having. While the formulation of the suggested actions (including a "why" section highlighting the impact), the progress visualization, and digital rewards on the platform were meant to support a feeling of impact, the interview data made clear that these features were not prominent or understood enough to be effective. Furthermore, participants described how they had forgotten to interact with the platform, even though they had planned to. As the platform had to be accessed via web browser and was otherwise designed to be unobtrusive, it was easy for a week to pass without participants being reminded of the platform. Finally, interview participants described issues of personalization, where receiving only three actions meant it was sometimes difficult to find actions, which could easily be completed in the location and with the amount of time and energy at their disposal in that moment.

Based on these findings, we propose in the following a list of ways in which technology for prosocial behavior change can be improved upon in the future.

## Implications for the design of effective technology for prosocial behavior change

**Design for actions to feel impactful for the user.** It is vital that users understand that their actions have an impact. While believing in one's capabilities to help in an everyday

context is important and should be examined, our results suggest that this may not be enough to invoke action. As past research [17, 19] has discussed and this study supports in new, it is crucial that potential helpers are aware not only of another person's need but also the ways in which their actions can have an impact. In this study, participants were reluctant to donate as donations tended to be considered a less meaningful and effective form of prosocial behavior. This is particularly relevant, as donations have in the past been used as a measure for prosocial behavior [41, 69]. These findings suggest that the processes leading to donations may be different from those involved in other forms of daily prosocial behavior.

Illustrating the impact an action will have is of course a challenge, as many social issues are often opaque in their causes and slow to be improved [70]. In this study, the focus was on prosocial actions, which could be undertaken in the local community. While at this level as well, change may not be easy and thereby also difficult to highlight, it may nevertheless offer some opportunity to show change—both individual and collective.

In this study, we used a map and badges to both reward participants and visualize progress, however these features were not made clear enough to participants and their benefit not evident enough. This highlights the importance of user-centered design and iterative improvement of features throughout the process of the creation of new technology [71]; Especially those designed for behavior change. Improved upon, such features may offer a possibility to make clear to users what impact they are having through their actions.

**Design for incorporation into everyday life.** While it may be enticing to believe otherwise, behavior change technology aiming to promote prosocial behavior is unlikely to become a central part of a user's daily life. Such technology will have to compete for the user's time with well-established habits and responsibilities [29, 72, 73], as well as countless other forms of technology [73]. In order to win over some of this time, our findings suggest that actions need to be easy to incorporate into existing daily routines as well as clearly valuable to the specific user. One approach here would be to ensure personalization of tasks is advanced enough that actions are suggested that the user will find fun and meaningful and can easily be performed with the time and at the place where the users is at that particular moment. An additional option would be to allow the user more choice, by offering a larger overview of potential actions, through which the user could then search and choose as they see fit.

Beyond this, our results point to the importance of ensuring that the platform is readily available and remains easily visible (e.g. by reminding users with notifications). In his work, Fogg [2, 74] describes the potential of ubiquitous technology to remind users to perform an action at the opportune moment when they are both motivated and capable of performing that action. For example, users could be reminded shortly before leaving the house in the morning to pack some spare change to give to anyone asking for a little money or to pack a reusable water bottle or coffee mug to prevent having to buy a one-way use alternative during the day. Fogg [2], however, also warns that if reminders are sent at a time when either motivation and capability are lacking, they will be perceived as annoying or frustrating rather than helpful. Identifying the correct time and place at which to send reminders is therefore challenging but an invaluable avenue for future improvement.

**Design for sustained usage over time.** While interacting with a new form of technology may be fun and novel the first time it is performed, a lack of variety may lead to loss of interest fairly quickly [48, 75]. Our results emphasize the importance of making the experience with the platform enjoyable. Even if the subject matter is serious and the motivation of the users to interact with the platform may be altruistic, as these results show, enjoying interacting with the platform is important to encourage users to want to return and engage with the platform over time. Enjoyment could, for example, be supported by introducing new features or narrative elements over time, a strategy long used effectively in games (e.g., [76, 77]). Another option

would be the use of social features, allowing users to hold each other accountable, motivate and support each other, or even join forces to completed actions [9, 78].

## Limitations and further research

While this study was strengthened by an experimental, theory-driven, longitudinal design, there are limitations, which we wish to discuss, both in terms of the limits to the interpretability of the results as well as in terms of avenues we see for future research.

This study advertised itself to potential participants explicitly as focusing on increasing prosocial behavior. This was done in order to attract participants interested in increasing their prosocial behavior (and detracting participants who would find this of no interest). However, we did not explicitly measure participants' motivation for participating. Thereby, it is uncertain what role motivation played in the effectiveness of the platform, as well as the degree to which these findings can be generalized to less interested populations. It is however important to note here that the goal of this platform was to work with participants own motivation to facilitate changes in their behavior. Extrapolations from these results onto ways of changing the behavior of people uninterested to do so, is therefore neither recommended, nor intended.

The use of a self-developed platform was decided on due to the lack of real-world platforms for prosocial action personalization and meant that we were flexible in measurement and stimuli manipulation. However, the simpleness of the platform also meant that, for example, the personalization was based on a pre-study assessment of motivation, capability and opportunity. As we learned, context- and time- sensitive understandings of the COM-B model is crucial in the context of prosocial behavior. Future research would therefore be advised to ensure that personalization is more sophisticated than that which was possible with this platform. For example, future research could improve the present methodology by providing real-time notifications based on the location of participants when they are near where they could accomplish a action.

The measure of daily prosocial behavior represented both a valuable source of information and a weakness in terms of accurate measurement. Across groups, daily prosocial behavior decreased during the study. This may have been due to the fact that daily measurements included both a quantitative and a qualitative measure of the past day's prosocial behavior. Most likely, participants fatigued particularly of the qualitative question, which necessitated them to write a list of prosocial actions. While the measure allowed the tracking of an approximate of prosocial behavior over time and across groups, future measurements should consider using only a quantitative, easy-to-complete measure, if asking for daily participant answers.

The platform was only accessible via the browser on either a laptop or a smartphone. Since multiple steps were needed to open the platform, log in and look for further assigned actions, this might have had an impact on accomplishing the assigned actions. It is not clear whether participants that did not use the platform, did so because they forgot to log their actions, or whether the platform design was too cumbersome and the effort too high for participants to log their actions. In terms of tracking daily prosocial actions, the timing of the notification (sent the next morning) might have lead to memory-loss-effects, that is, participants forgetting their prosocial actions by the time they were reminded to track them. Future research could improve this by increasing the frequency of the email reminders, although this could lead to annoyed participants.

The results of this study suggest that the relationship between self-efficacy and prosocial behavior is more nuanced than previously assumed—with the self-efficacy for performing concrete prosocial actions and self-efficacy for acting in a way which will have a tangible impact playing differing roles in the interplay between technology and behavior. The *Everyday*

*Helping Self-Efficacy* scale was developed in its current state in order to offer a first, more specific examination of self-efficacy in the context of everyday helping behavior. Considering the importance of self-efficacy for behavior, we would encourage the examination of this and other forms of specific self-efficacy in the context of prosocial behavior. Beyond this, examinations of the interplay between different forms of self-efficacy may also offer more nuanced views on the forces behind behavior.

While data collection over three weeks offered first insights into the relationships between the examined variables, a longer running time of the study—especially also the inclusion of data points after exposure to the platform and daily prosocial behavior tracking ceased—could offer future studies additional insights into the interaction between technological tools and behavior change over time.

## Conclusion

Designing for outcomes as diverse and complex as prosocial behavior may seem daunting. However, research on the interaction between technology and psychology can offer valuable insights into how to design to support prosocial behavior. The findings of this study point to the potentials and pitfalls of attempting to use one such form of technology to increase prosocial behavior over time. Tracking prosocial behavior over time was related to increases in participants' belief in their ability to help others in everyday settings. However, in order to increase prosocial behavior, these results suggest that other features of the platform would need to be improved upon in order to support participants' belief in their ability to affect change. This could potentially be achieved by highlighting the impact of suggested actions and creating engaging progress visualisations. As such technology will need to compete for the user's time and attention, it is important to ensure that it fits well into everyday life and is prominent enough to not be forgotten. Personalizing to individuals capabilities, opportunities, and motivations, as well as sending reminders at moments when participants are both motivated and capable of taking action may offer avenues for achieving this. Finally, for continued usage over time, it is important to ensure that users are enjoying themselves. Ensuring variety in suggested actions and facilitating social interactions are possible ways in which this could be supported. While further studies are necessary to understand this research area in more detail, the exploratory findings of this study provide some first insights into ways in which technology can be designed to support prosocial behavior change over time. Together, progression in this research area will hopefully lead to a greater understanding of how we can design technology to benefit individuals, communities, and, ultimately, societies.

## Acknowledgments

Many thanks to Melanie Svab for her valuable support—her record-breaking speed at learning LaTeX wizardry helped get this manuscript out the door.

## Author Contributions

**Conceptualization:** Sharon T. Steinemann, Benjamin J. Geelan.

**Data curation:** Sharon T. Steinemann, Stephan Zaehringer, Kamalatharsi Mutuura, Ewgenij Wolkow.

**Formal analysis:** Sharon T. Steinemann, Stephan Zaehringer.

**Funding acquisition:** Sharon T. Steinemann.

**Investigation:** Sharon T. Steinemann, Stephan Zaehringer.

**Methodology:** Sharon T. Steinemann, Benjamin J. Geelan.

**Project administration:** Sharon T. Steinemann.

**Resources:** Sharon T. Steinemann, Benjamin J. Geelan, Kamalatharsi Mutuura, Ewgenij Wolkow, Klaus Opwis.

**Software:** Benjamin J. Geelan, Kamalatharsi Mutuura, Ewgenij Wolkow, Lars Frasseck.

**Supervision:** Sharon T. Steinemann, Klaus Opwis.

**Validation:** Sharon T. Steinemann.

**Visualization:** Sharon T. Steinemann.

**Writing – original draft:** Sharon T. Steinemann.

**Writing – review & editing:** Sharon T. Steinemann, Benjamin J. Geelan, Stephan Zaehringer, Kamalatharsi Mutuura, Ewgenij Wolkow, Lars Frasseck, Klaus Opwis.

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
