## [Decision Letter · Decision Letter 0]

14 Jan 2020

PONE-D-19-27014

Potentials and Pitfalls of Increasing Prosocial Behavior and Self-Efficacy over Time using an Online Personalized Platform

PLOS ONE

Dear Ms Steinemann,

Thank you for submitting your manuscript to PLOS ONE. After careful consideration, we feel that it has merit but does not fully meet PLOS ONE’s publication criteria as it currently stands. Therefore, we invite you to submit a revised version of the manuscript that addresses the points raised during the review process.

Your paper has been reviewed by two experts in the addressed field. Overall, one of the reviewers advises against the publication of the paper, considering different shortcomings of the manuscript, especially related to key factors such as the analyses performed (some key ones remain missing, some others need revision and better interpretations) and the quality and pertinence of the discussion given to the results. However, the second reviewer remarks some potential in the article and, as I agree in it, I would like to give the authors the opportunity to revise and resubmit the paper.

We would appreciate receiving your revised manuscript by Feb 28 2020 11:59PM. To enhance the reproducibility of your results, we recommend that if applicable you deposit your laboratory protocols in protocols.io, where a protocol can be assigned its own identifier (DOI) such that it can be cited independently in the future. For instructions see: http://journals.plos.org/plosone/s/submission-guidelines#loc-laboratory-protocols

We look forward to receiving your revised manuscript.

Kind regards,

Sergio A. Useche, Ph.D.

Academic Editor

PLOS ONE

Journal Requirements:

2. We note that Table 3 may include questionnaire items that may have been previously published. The reproduction of previously published has implications for the copyright that may apply to these publications. We would be grateful if you could clarify whether you have obtained permission from the original copyright holder to republish these items under a CC BY license. If you have not obtained permission to publish these items please remove them from your manuscript. You may wish to replace the text you have removed with relevant question numbers/ brief descriptions of each item.

3. Please ensure that you include a title page within your main document. You should list all authors and all affiliations as per our author instructions and clearly indicate the corresponding author.

Additional Editor Comments (if provided):

Reviewers' comments:

Reviewer's Responses to Questions

**Comments to the Author**

1. Is the manuscript technically sound, and do the data support the conclusions?

Reviewer #1: Yes

Reviewer #2: No

2. Has the statistical analysis been performed appropriately and rigorously? 

Reviewer #1: Yes

Reviewer #2: No

3. Have the authors made all data underlying the findings in their manuscript fully available?

Reviewer #1: Yes

Reviewer #2: Yes

4. Is the manuscript presented in an intelligible fashion and written in standard English?

Reviewer #1: Yes

Reviewer #2: Yes

5. Review Comments to the Author

Reviewer #1: This is a well-written article on the effect of using a digital platform in prosocial behaviors. The theoretical background of the work clearly lays out each component of the research. However, the two studied measures of self-efficacy (change impact and everyday helping) could have been introduced early on.

The authors present a detailed description of the study design and developed platform components. As for the platform, it is stated that the goal of the research was to create a platform where participants can successfully master prosocial actions. It is not clear, in designing the platform, how or why the critical components of the platform, such as personalized actions, gamified elements, and virtual rewards were selected. A strong justification for including these components in the platform would strengthen the article.

Similarly, while details on the personalization of suggested actions were given, such details for the other two components (gamification and virtual rewards) are lacking but critical for comprehending the merit of the work.

Furthermore, information on how the COM-B model maps with these components would make the article stronger. For example, how is the capability, opportunity, and motivation of the users are known and integrated with the platform? Are these dynamic or static? Are these measures context-sensitive? Do these changes based on user activities over time?

In the study design, three variables (enjoyment, appreciation, usability) are measured to assess platform effectiveness. The theoretical foundation or justification for using these variables should be included.

A rigorous data analysis section adds value to the article. Adequate discussion is presented to share detailed insight. Limitations and further research address some of the critical aspects of the research. In the Limitations section, authors should consider addressing how the design or execution of the platform may or may not have affected the findings. The team developed its own digital platform to carry out the research. It would be interesting to know if the design and implementation of the platform played any role in the findings.

As part of the research, the team developed a scale to measure everyday helping self-efficacy. This is a valuable contribution. Design implications for fostering prosocial behavior through a digital platform is another valuable contribution of the work.

Reviewer #2: This paper randomizes 66 psychology students (who is this case are experimental subjects who are interested in increasing their prosocial behavior) into two groups, Baseline and Platform. All subjects were asked to self-report their prosocial behavior for 3 weeks (daily measures and weekly), while subjects in the Platform treatment also had access to an online platform that recommends, gamifies, and rewards action. The main theoretical framework was that of self-efficacy - the belief that one's ability to successfully perform an action - will increase the likelihood that a behavior will be taken. The findings is that the platform has no effect on either the behavior or self-efficacy. If done convincingly, this is an interesting null finding and could be important, but there are other issues that weaken this study.

First, the paper does not discuss self-efficacy in a consistent way. For example believing that I have the discipline to put aside my money to donate is different from believing that the contribution will make a difference(p.2), enjoying the activity more (top of p.3), or developing more empathy (bottom of p.3). The connections between change impact self efficacy (p. 8), everyday helping (p.9) and general self efficacy (p.10) are not well developed. It is unclear whether including two of these measures result in multicollinearity (e.g in Table 6), and whether we expect a change in self efficacy to result in a change in behavior (which suggest a different econometric model than what is employed). I would suggest that the paper be refocused on self-efficacy, because right now the focus is too diffused.

Second, I think there is just not enough data (or theoretical backing) to discuss Likelihood to use. We have 33 people on the platform, with several dropped for being an outlier or not completing the survey, and we are splitting this further to Likely to use and Not Likely to use. On the topic of not enough data, we have 8 people in post-interview, which does not feel adequate for the claims that this paper is making, especially since the qualitative and quantitative results are different.

6. PLOS authors have the option to publish the peer review history of their article (what does this mean?). If published, this will include your full peer review and any attached files.

Reviewer #1: Yes: Monika Akbar

Reviewer #2: No

---

## [Author Response · Author response to Decision Letter 0]

14 Apr 2020

Dear editor, dear reviewers,

We would like to thank you for your constructive, careful and helpful feedback. We have tried to address all points made and believe the final manuscript to be stronger as a consequence of it. The greatest changes to the manuscript are the removal of the results and discussion of likelihood to use, an earlier and more thorough examination of self-efficacy theory, more detailed linking between theory and platform design, and a critical examination of the connection between platform design and research findings. 

In the following are the point-by-point responses to all review points made.

Reviewer #1 Points

Point 1

- Reviewer 1: This is a well-written article on the effect of using a digital platform in prosocial behaviors. The theoretical background of the work clearly lays out each component of the research. However, the two studied measures of self-efficacy (change impact and everyday helping) could have been introduced early on.

- Author Response: The introduction has been revised to introduce the two measures of self-efficacy early on.

- Corresponding sections and page number(s) in the revised manuscript: Introduction (p. 3 in the revised manuscript)

Point 2

- Reviewer 1: The authors present a detailed description of the study design and developed platform components. As for the platform, it is stated that the goal of the research was to create a platform where participants can successfully master prosocial actions. It is not clear, in designing the platform, how or why the critical components of the platform, such as personalized actions, gamified elements, and virtual rewards were selected. A strong justification for including these components in the platform would strengthen the article.

- Author Response: A point-by-point justification for each included component (personalized actions and gamified elements, including virtual rewards) has been added to the section describing the platform.

Introduction

- Corresponding sections and page number(s) in the revised manuscript: (p. 5 in the revised manuscript)

Point 3

- Reviewer 1: Similarly, while details on the personalization of suggested actions were given, such details for the other two components (gamification and virtual rewards) are lacking but critical for comprehending the merit of the work.

- Author Response: Details on the gamification (virtual rewards and map) have been added.

Method

- Corresponding sections and page number(s) in the revised manuscript: (p. 7-9 in the revised manuscript)

Point 4

- Reviewer 1: Furthermore, information on how the COM-B model maps with these components would make the article stronger. For example, how is the capability, opportunity, and motivation of the users are known and integrated with the platform? Are these dynamic or static? Are these measures context-sensitive? Do these changes based on user activities over time?

- Author Response: Information on how the COM-B model mapped to the components of the platform, including information on how we gained information on participants capability, opportunity, and motivation has been added. A discussion of the lack of context sensitivity has been added to the Limitation section.

- Corresponding sections and page number(s) in the revised manuscript: Introduction (p. 7-9 in the revised manuscript) & Discussion (p. 25 in the revised manuscript)

Point 5

- Reviewer 1: In the study design, three variables (enjoyment, appreciation, usability) are measured to assess platform effectiveness. The theoretical foundation or justification for using these variables should be included.

- Author Response: The theoretical foundation for enjoyment, appreciation, and usability has been added in the Platform experience section.

- Corresponding sections and page number(s) in the revised manuscript: Method (p. 12 in the revised manuscript)

Point 6

- Reviewer 1: A rigorous data analysis section adds value to the article. Adequate discussion is presented to share detailed insight. Limitations and further research address some of the critical aspects of the research. In the Limitations section, authors should consider addressing how the design or execution of the platform may or may not have affected the findings. The team developed its own digital platform to carry out the research. It would be interesting to know if the design and implementation of the platform played any role in the findings

- Author Response: A discussion of limitations created by the design and execution have been added and how the design and implementation played a role in the findings.

- Corresponding sections and page number(s) in the revised manuscript: Discussion - Limitations (p. 25 in the revised manuscript)

Point 7

- Reviewer 1: As part of the research, the team developed a scale to measure everyday helping self-efficacy. This is a valuable contribution. Design implications for fostering prosocial behavior through a digital platform is another valuable contribution of the work.

- Author Response: Further discussion of the design implications have been added to the Discussion.

- Corresponding sections and page number(s) in the revised manuscript: Discussion (p. 23-25 in the revised manuscript)

Reviewer #2

Point 8

- Reviewer 2: This paper randomizes 66 psychology students (who is this case are experimental subjects who are interested in increasing their prosocial behavior) into two groups, Baseline and Platform. All subjects were asked to self-report their prosocial behavior for 3 weeks (daily measures and weekly), while subjects in the Platform treatment also had access to an online platform that recommends, gamifies, and rewards action. The main theoretical framework was that of self-efficacy - the belief that one's ability to successfully perform an action - will increase the likelihood that a behavior will be taken. The findings is that the platform has no effect on either the behavior or self-efficacy. If done convincingly, this is an interesting null finding and could be important, but there are other issues that weaken this study. First, the paper does not discuss self-efficacy in a consistent way. For example believing that I have the discipline to put aside my money to donate is different from believing that the contribution will make a difference(p.2), enjoying the activity more (top of p.3), or developing more empathy (bottom of p.3). The connections between change impact self efficacy (p. 8), everyday helping (p.9) and general self efficacy (p.10) are not well developed.

- Author Response: The discussion of self-efficacy has been revised to explain the nuances between these sections and how they tie into overarching self-efficacy theory.

- Corresponding sections and page number(s) in the revised manuscript: Introduction (p. 2-3 in the revised manuscript)

Point 9

- Reviewer 2: It is unclear whether including two of these measures result in multicollinearity (e.g in Table 6)...

- Author Response: Centering the data, as we did, is used to stabilize and help prevent issues with multicollinearity in multilevel analysis (Field, Miles & Field, 2012). We have now rerun the analysis with the variables separately to compare results to the combined model, which did not lead to different results. A note on this is now included in the manuscript.

- Corresponding sections and page number(s) in the revised manuscript: Results (p. 17 in the revised manuscript)

Point 10

- Reviewer 2: ... and [ It is unclear] whether we expect a change in self efficacy to result in a change in behavior (which suggest a different econometric model than what is employed). I would suggest that the paper be refocused on self-efficacy, because right now the focus is too diffused.

- Author Response: The theoretical background of self-efficacy and it’s close relation to behavior has been further emphasized. As described in previous points, the centrality of self-efficacy has now been further set into focus.

- Corresponding sections and page number(s) in the revised manuscript: Introduction (p. 2-3 in the revised manuscript)

Point 11

- Reviewer 2: Second, I think there is just not enough data (or theoretical backing) to discuss Likelihood to use. We have 33 people on the platform, with several dropped for being an outlier or not completing the survey, and we are splitting this further to Likely to use and Not Likely to use.

- Author Response: We agree that the sample discussed in the section on Likelihood of Use is small, but believe that Likelihood to Continue is important to discuss. We have therefore removed the results and discussion of the subsample analysis and refer only to the correlational results introduced in the first section on the Results. 

- Corresponding sections and page number(s) in the revised manuscript: Results and Discussion (p. 18, p. 22- 26 in the revised manuscript)

Point 12

- Reviewer 2: On the topic of not enough data, we have 8 people in post-interview...

- Author Response: The theoretical foundation behind the qualitative data analysis has been expanded to explore the reasoning behind the sample size chosen, which is in line with recommended sample sizes for such an interview analysis.

- Corresponding sections and page number(s) in the revised manuscript: Results (p. 14-15 in the revised manuscript)

Point 13

- Reviewer 2: ...which does not feel adequate for the claims that this paper is making, especially since the qualitative and quantitative results are different.

- Author Response: The discussion on the relationship between the quantitative and qualitative results has been extensively rewritten to reflect the synergistic potential of these methods. 

- Corresponding sections and page number(s) in the revised manuscript: Discussion (p. 22-23 in the revised manuscript)

Journal Requirements

Point 14

- Journal Requirements: Please ensure that your manuscript meets PLOS ONE's style requirements, including those for file naming. The PLOS ONE style templates can be found at http://www.journals.plos.org/plosone/s/file?id=wjVg/PLOSOne_formatting_sample_main_body.pdf and http://www.journals.plos.org/plosone/s/file?id=ba62/PLOSOne_formatting_sample_title_authors_affiliations.pdf

- Author Response: The LaTeX Template provided on the PLOS ONE website was used and modified based on the provided style guide. Please let us know if other changes are requested.

- Corresponding sections and page number(s) in the revised manuscript: Overall

Point 15

- Journal Requirements: We note that Table 3 may include questionnaire items that may have been previously published. The reproduction of previously published has implications for the copyright that may apply to these publications. We would be grateful if you could clarify whether you have obtained permission from the original copyright holder to republish these items under a CC BY license. If you have not obtained permission to publish these items please remove them from your manuscript. You may wish to replace the text you have removed with relevant question numbers/ brief descriptions of each item.

- Author Response: All items were generated through the pilot study in this survey and have not been published in any previous publications. This has been added to the manuscript

- Corresponding sections and page number(s) in the revised manuscript: Method (p. 12 in the revised manuscript)

Point 16

- Journal Requirements: Please ensure that you include a title page within your main document. You should list all authors and all affiliations as per our author instructions and clearly indicate the corresponding author.

- Author Response: The title page with all authors, affiliation and corresponding author are now included.

- Corresponding sections and page number(s) in the revised manuscript: Title page (p. 1 in the revised manuscript)

Point 17

- Journal Requirements: Please include captions for your Supporting Information files at the end of your manuscript, and update any in-text citations to match accordingly. Please see our Supporting Information guidelines for more information: http://journals.plos.org/plosone/s/supporting-information.

- Author Response: Captions for the supporting information files have been added to the end of the manuscript.

- Corresponding sections and page number(s) in the revised manuscript: Supporting Information (p. 32 in the revised manuscript)

References

Field, A., Miles, J., & Field, Z. (2012). Discovering statistics using R. Sage publications.

---

## [Decision Letter · Decision Letter 1]

27 May 2020

Potentials and pitfalls of increasing prosocial behavior and self-efficacy over time using an online personalized platform

PONE-D-19-27014R1

Dear Dr. Steinemann,

We are pleased to inform you that your manuscript has been judged scientifically suitable for publication and will be formally accepted for publication once it complies with all outstanding technical requirements.

With kind regards,

Sergio A. Useche, Ph.D.

Academic Editor

PLOS ONE

Additional Editor Comments (optional):

Reviewers' comments:

Reviewer's Responses to Questions

**Comments to the Author**

1. If the authors have adequately addressed your comments raised in a previous round of review and you feel that this manuscript is now acceptable for publication, you may indicate that here to bypass the “Comments to the Author” section, enter your conflict of interest statement in the “Confidential to Editor” section, and submit your "Accept" recommendation.

Reviewer #2: All comments have been addressed

2. Is the manuscript technically sound, and do the data support the conclusions?

Reviewer #2: Yes

3. Has the statistical analysis been performed appropriately and rigorously? 

Reviewer #2: Yes

4. Have the authors made all data underlying the findings in their manuscript fully available?

Reviewer #2: Yes

5. Is the manuscript presented in an intelligible fashion and written in standard English?

Reviewer #2: Yes

6. Review Comments to the Author

Reviewer #2: (No Response)

7. PLOS authors have the option to publish the peer review history of their article (what does this mean?). If published, this will include your full peer review and any attached files.

Reviewer #2: No

---

## [Editor Report · Acceptance letter]

15 Jun 2020

PONE-D-19-27014R1 

Potentials and pitfalls of increasing prosocial behavior and self-efficacy over time using an online personalized platform 

Dear Dr. Steinemann:

I'm pleased to inform you that your manuscript has been deemed suitable for publication in PLOS ONE. Congratulations! Your manuscript is now with our production department. 

Kind regards, 

on behalf of

Dr. Sergio A. Useche 

Academic Editor

PLOS ONE